# Shaping contactless radiation forces through anomalous acoustic scattering

Matthew Stein[1], Sam Keller[1], Yujie Luo ®[1] & Ognjen Ilic ®[1] ✉

Waves impart momentum and exert force on obstacles in their path. The transfer of wave momentum is a fundamental mechanism for contactless manipulation, yet the rules of conventional scattering intrinsically limit the radiation force based on the shape and the size of the manipulated object. Here, we show that this intrinsic limit can be broken for acoustic waves with subwavelength-structured surfaces (metasurfaces), where the force becomes controllable by the arrangement of surface features, independent of the object's overall shape and size. Harnessing such anomalous metasurface scattering, we demonstrate complex actuation phenomena: self-guidance, where a metasurface object is autonomously guided by an acoustic wave, and tractor beaming, where a metasurface object is pulled by the wave. Our results show that bringing the metasurface physics of acoustic waves, and its full arsenal of tools, to the domain of mechanical manipulation opens new frontiers in contactless actuation and enables diverse actuation mechanisms that are beyond the limits of traditional wave-matter interactions.

The process of momentum exchange between a wave and an object underpins radiation forces exerted over liquid, gaseous and solid matter[1]. For macroscale objects, applications of wave forces have led to actuation mechanisms such as acoustic handling and transport[2–8], spawning interest as promising tools across biology and biomedicine to chemistry and colloidal science[9–17]. However, to date, such mechanisms have relied on conventional laws of wave refraction that govern how a wave interacts with the object it manipulates. For a typical surface or interface, the transfer of momentum is governed by Snell's law for waves, which means that the wave force is dictated and constrained by the overall shape and size of the object. In turn, this reliance fundamentally constrains the kind of forces that can be induced and therefore limits the range of actuation behaviors that can be generated. For example, common actuation configurations work only for subwavelength objects[18–20] or objects of particular symmetric shape[21–23] where actuating energy potentials are easy to create. Further, efforts to overcome such limitations have relied on complex active feedback control, typically employing some form of adaptive adjustment of acoustic fields, often in conjunction with object tracking[24,25]. At its root, the challenge lies in the entangled relationship between the shape of the object (which dictates scattering) and its mechanical response to waves (which is dictated by scattering).

Here, we demonstrate that patterning the object's surface into a metasurface—an embedded array of subwavelength elements—can overcome such wave force limitations by means of localized control of wave refraction. Metasurfaces are attracting significant interest due to their remarkable capability to synthesize complex wavefronts[26–33], with diverse applications in beam-steering, cloaking, and wave focusing[34–43]. Similar to optical manipulation[44–46], the ability to steer acoustic waves can also be seen as a mechanical phenomenon, and the underlying metasurface physics can be harnessed to shape mechanical forces. Unlike conventional wave refraction, a metasurface object exhibits anomalous refraction that is expressed through the generalized Snell's law, $\sin\theta_R = \sin\theta_I + \frac{1}{k}\frac{d\Phi(\mathbf{r})}{dr}$, where the relationship between the refracted angle ($\theta_R$) and the incident angle ($\theta_I$) is augmented by the local phase term $\Phi(\mathbf{r})$ at the position $\mathbf{r}$, as shown in Fig. 1 (here, $k$ is the wave-vector). Since the relationship between the refracted and incident angle reflects the balance between the outgoing and the incoming wave momentum, the control of the local phase $\Phi(\mathbf{r})$ thus translates into the control of the local force $\mathbf{F}(\mathbf{r})$ that is exerted on the metasurface object. Therefore, the purposeful arrangement of metasurface unit cells on the subwavelength scale waves becomes a new way to realize desired actuation behavior on the large (object) scale.

[1]Department of Mechanical Engineering, University of Minnesota, Minneapolis, MN 55455, USA. ✉e-mail: ilic@umn.edu

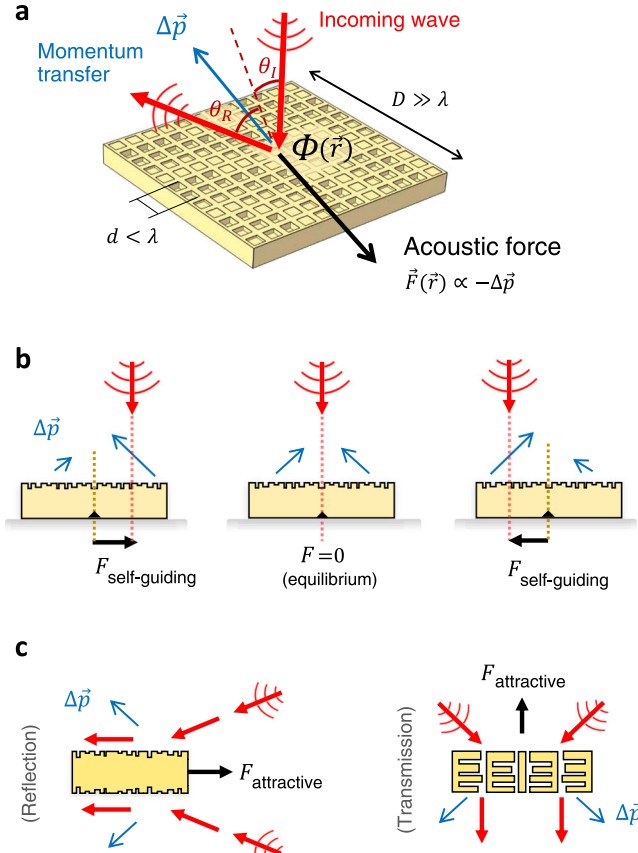

**Fig. 1 | Shaping contactless forces with metasurfaces that steer acoustic waves.** **a** The local force **F**(**r**) is proportional to the momentum change Δ**p** (blue), which is controlled by the local phase $\Phi(\mathbf{r})$. Through the spatial design of the subwavelength ($d < \lambda$) unit cells of the metasurface, complex force profiles can be generated to enable new mechanisms for contactless actuation. **b** Self-guiding metasurfaces can autonomously self-lock to and follow the remote wave source. Their ability to move as the source moves stems from the scattering asymmetry which pins the center of the metasurface to the source radiation axis (dashed red). E.g., when the metasurface is aligned on the axis, no net force is present (middle); but if the source has moved the metasurface will follow (left, right). **c** Pulling force: metasurfaces can be not just pushed from, but also pulled towards the radiation source (here shown as two independent non-interacting incident waves).

## Results

We focus on metasurfaces as promising platforms for shaping mechanical actuation because they can endow an object with acousto-mechanical functionality that is independent of, and unconstrained by its morphology or dimensions. To demonstrate this capability, in this work we synthesize three metasurface objects that are seemingly identical (all are rectangular blocks) but exhibit three different actuation functionalities. First, we show a metasurface whose subwavelength unit cells are arranged to enable a strong lateral wave momentum shift, leading to a dominant sideways force that is parallel to the object's surface. This stands in contrast to conventional refraction, where the only force that is exerted is normal to the surface. Second, we show a metasurface with self-guiding capability: it can sense the change in the acoustic field around it and respond to the change in an autonomous manner (Fig. 1b). Specifically, the metasurface is able to lock itself to the remote acoustic source: when the source is moved, the metasurface object follows the path. Third, we show tractor beam-like behavior where a metasurface is contactlessly pulled towards the source of radiation (Fig. 1c). It is noteworthy to point out that all three of these phenomena (a strong lateral force, self-guiding, pulling) are typically not possible for such

rectangular blocks, as conventional scattering cannot provide the necessary momentum balance. But here such shape constraints are removed−in fact, apart from the different patterns of subwavelength features, the overall shape of the metasurface is the same in all cases in this work.

### Acoustic force controlled by metasurface scattering

To probe the connection between the subwavelength arrangement of metasurface unit cells and the radiation pressure it experiences, we first establish a predictive metasurface force model. Intuitively, a metasurface that steers sound waves in a particular direction should experience a force opposite to the change in the wave momentum Δ**k** (Fig. 1). To obtain a structure-to-force mapping that will guide our metasurface designs, we employ a computational approach to calculate the second-order pressure terms. We numerically estimate the radiation force on a metasurface as the closed contour integral of the second-order acoustic components, i.e., $\mathbf{F} = -\int_S [\langle p_2 \rangle \boldsymbol{n} + \rho_0 \langle (\boldsymbol{n} \cdot \mathbf{v}_1)\mathbf{v}_1 \rangle] dA$, where $\mathbf{v}_1, p_2$ are the first and second-order perturbations in velocity and pressure, respectively, and $\rho_0$ is the density of the medium[47,48]. The integral can be carried out along any fixed surface $S$ enclosing an object, as ensured by momentum conservation. We validate this approach (see the Supplementary Materials) and use it to infer the direction and the magnitude of the acoustic force on a metasurface. We use this framework to guide the design of the metasurface−i.e., to select the arrangement of its unit cells −for each of the three target mechanical applications discussed in this work.

To demonstrate a proof-of-concept of contactless actuation of a metasurface, our first target mechanical behavior is a design that exhibits a strong lateral force. We select an inaudible operating frequency (20 kHz) for noiseless demonstration and to minimize disruptions. Figure 2a shows the building-block unit cell consisting of a U-shaped structure of subwavelength width. We chose this unit cell shape for its simplicity of fabrication, but we note that the concept and the phenomena that we report are general and not unique to the chosen unit cell design (in the Supplementary Information, we also investigate a metasurface with a space-coiling unit cell topology to demonstrate the same physics). The depth of the grooves ($x_i$) determines the local scattered phase, and we map the relationship between the depth and the phase through finite-element simulations of acoustic propagation in COMSOL Multiphysics (Fig. S3). Arranging the unit cells based on the depth-phase relationship, we can achieve a sideways wavefront shift and, further, we can refine the topology of the metasurface to enhance the metasurface force. Specifically, we can treat the depths of the $N$ unit cells ($x_1, \ldots, x_N$) as varying, but independent design parameters for an iterative optimization process where each step $\mathbf{F}(x_1, \ldots, x_N)$ is evaluated through finite-element analysis (see the "Methods" section).

The outcome of this process is a metasurface shown in Fig. 2a (bottom), comprising $N = 30$ unit cells and the overall dimensions of length = 77.7 mm, width = 20.2 mm, and height = 11.1 mm. For scale, a US dime is also shown. The description of the metasurface and its dimensions is presented in the Supplementary Materials. Figure 2b shows the scattered wave profile obtained from finite-element analysis in COMSOL Multiphysics. Radiation is normally incident on the metasurface from a set of piezoelectric transducers (operating at $f = 20$ kHz, $\lambda = 17$ mm in the air), to mimic the experimental configuration which is discussed below. We observe strong asymmetric scattering to the right, which is a sign that the metasurface should be experiencing a lateral push to the left. Interestingly, the side-scattering, while strong, exhibits an angular spread. In a conventional steering/focusing application, such an angular spread might be detrimental and undesirable in a metasurface. But for mechanical actuation, the key metric is the force; as long as the desired force profile is maintained, it is useful that a metasurface can in fact tolerate and be robust to angular spreading in scattering.

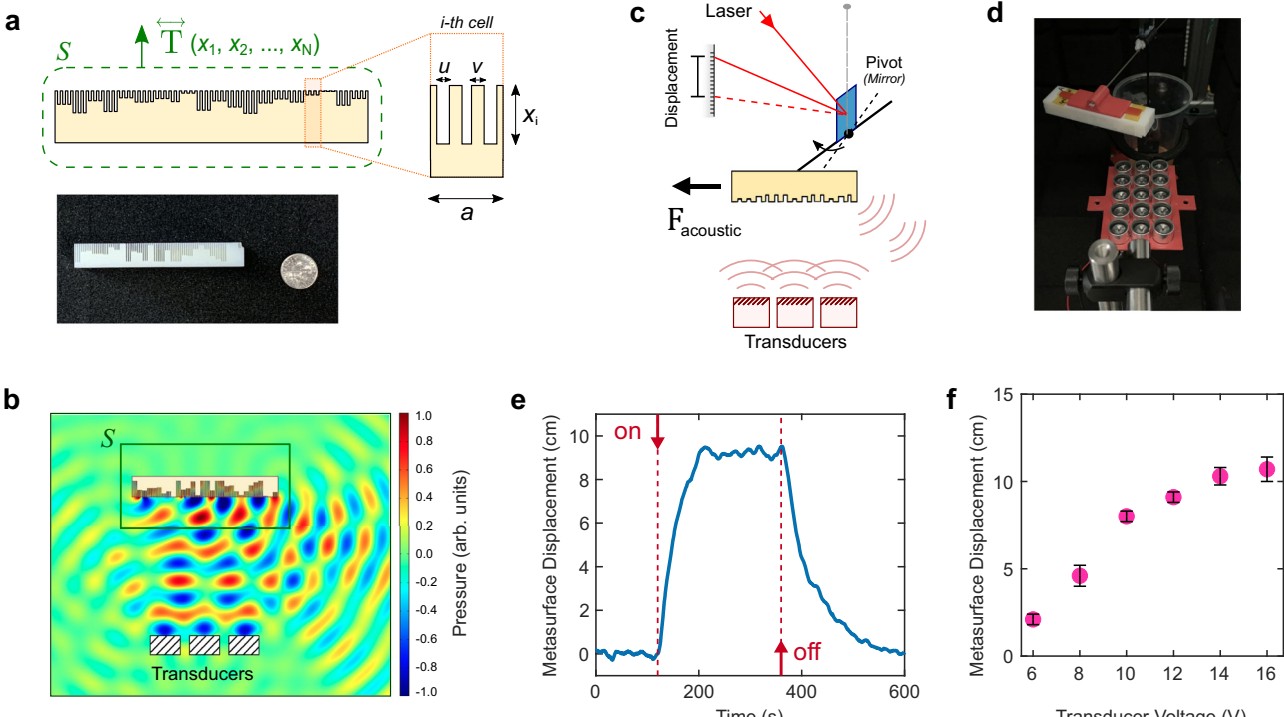

**Fig. 2 | Design of the acousto-mechanical metasurface and the demonstration of actuation. a** The building-block unit cell with variable depth (inset). The metasurface is optimized to induce a strong lateral force parallel to its long side, as derived from the pressure tensor **T**. Below: fabricated metasurface (a dime is shown for scale). **b** Scattering profile for the incident inaudible acoustic wave (20 kHz) showing the momentum flow to the right. Box indicates the enclosing surface over which the pressure tensor is numerically integrated. **c** Schematic of the setup to measure metasurface actuation (see Fig. S1 for details). **d** Photo of the metasurface when actuated (Supplementary Movie 2). **e** Metasurface displacement versus time. Dashed lines denote the on/off turning off sound source. **f** Metasurface equilibrium displacement versus transducer voltage. Error bars represent standard deviation.

Our proof-of-concept experimental demonstration of metasurface actuation is displayed in Fig. 2c, d. In the experiment, the metasurface is oriented with its unit cells facing a set of piezoelectric transducers operating at 20 kHz. The setup developed to measure actuation due to acoustic waves is based on the torsion-pendulum concept (details in the SI, e.g., Fig. S1 shows the rendition of the setup). When a wave anomalously refracts off of a metasurface, a lateral force is induced, and the pendulum deflects in response. This motion is observed as the displacement of the laser spot from a laser that is reflected off the mirror at the pendulum base (Fig. 2c). A camera feed of the screen provides live tracking of metasurface actuation. Simultaneously, a spot-tracing algorithm detects the laser spot in real time and translates this information into the position of the metasurface. Initially, the metasurface is not actuated. As soon as sources are turned on, a sharp jump in the equilibrium position is observed (Fig. 2e, dashed red line labeled "on"); once the sources of sound are turned off, the metasurface returns to its initial equilibrium position. See Supplementary Movies 1 and 2 for details. The photo in Fig. 2d shows the side view of the actuated metasurface. Crucially, the direction of the movement aligns with the predicted direction of the radiation force, indicating that the metasurface is indeed actuated as designed.

We characterize the strength of the actuation as a function of the signal supplied to the transducers. Figure 2f displays the change in the metasurface equilibrium position when the voltage is varied. A theoretical model predicts that the displacement will scale with the force as described by the relationship $F = \frac{\eta R^4 \pi}{2hL}\psi$, where $\psi$ is the torsional rotation angle, $R$ is the radius of the tungsten fiber, $\eta$ is the shear modulus of the fiber, and $h,L$ are the length of the fiber and the length of the pendulum arm, respectively (see the Supplementary Materials). The data in Fig. 2f confirm the trend of stronger force with increased transducer voltage, but also show a deviation from the expected

behavior that is more pronounced at higher source voltages/powers. We primarily attribute this discrepancy to the fact that the metasurface, when actuated, is noticeably displaced relative to the acoustic source and thus not intercepting the full acoustic wave at normal incidence—as can be seen in Fig. 2d. An additional source of discrepancy is the intrinsic variability of phase and intensity profile among the transducers. In the experiment, the orientation of the source, the metasurface, and the rotating arm are intentionally chosen to ensure that no parasitic forces can affect deflection (see Fig. S1). Finally, a measurement with a flipped metasurface yields the same behavior but with the opposite sign of deflection, eliminating the possibility that the observed effect is an experimental artifact, and further confirming that the origin of the force is the metasurface itself.

## Metasurface self-guiding

Having established the proof-of-concept of metasurface actuation, we proceed to introduce and demonstrate the self-guiding mechanism. A metasurface that guides itself must be capable of responding to changes in the acoustic intensity in its vicinity. Figure 3a shows the configuration where the source—comprising a set of transducers—is allowed to move freely. The curved orientation of the transducers is chosen to define a radiation axis that sets the equilibrium position of the metasurface. To probe the self-guiding phenomenon, we design a metasurface with a center-symmetric arrangement of unit cells. When the metasurface is aligned with the axis of the radiation source, the scattering is symmetric, and no lateral force is present. However, when the source is offset, a net force is induced due to the asymmetry in scattering. The fabricated metasurface, shown in Fig. 3a (right), is designed to drive itself back to its on-axis equilibrium position. For scale, a US dime is also shown. The description of the metasurface and

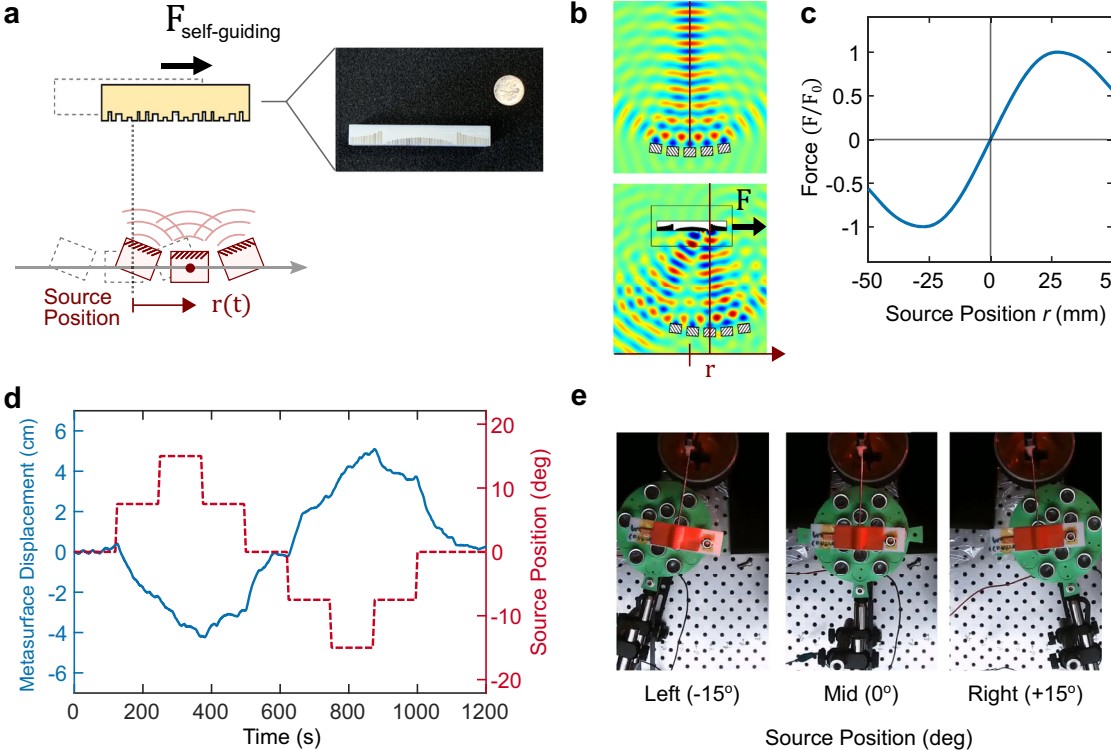

**Fig. 3 | Metasurface self-guiding: concept and demonstration. a** Metasurface is designed to autonomously track and guide itself to follow the movement of the radiation source. Right: fabricated metasurface (a dime is shown for scale). **b** Top: Focused acoustic intensity profile emanating from a set of piezoelectric transducers that comprise the source of radiation (20 kHz). The solid line denotes the radiation axis. Bottom: Scattering map for a case when the metasurface is displaced away from the radiation axis. **c** Force as a function of the source position. The shape of the curve indicates guiding behavior: force is positive for positive source position, and vice-versa. **b**, **c** are simulation results. **d** Experiment: the metasurface position (blue) tracks the position of the radiation source (red), as a function of time. **e** Top view of the camera frames at several instances during actuation (Supplementary Movie 3).

the transducer arrangement is presented in the Supplementary Materials.

Figure 3b displays the simulated acoustic intensity profile without (top) and with (bottom) the metasurface in the computational domain. In the absence of the metasurface, the radiation source exhibits a focused profile with a defined radiation axis (solid line, Fig. 3b top). When the metasurface is introduced and the source is offset to the right, we observe strong refraction sideways to the left. This indicates that the metasurface would experience a force in the opposite direction (to the right), thus tracking the source. Figure 3c plots the simulated lateral force on the metasurface as a function of the source position. The force is normalized to the maximum force achieved at the source displacement of ≈25 mm. The shape of the force curve confirms guiding behavior: the force on the metasurface is positive for the positive source position, and vice-versa.

To experimentally demonstrate the self-guiding capability, we characterize the motion of the metasurface in response to a moving source. Figure 3d displays the pre-programmed position of the radiation source as a function of time (red dashed line), and the corresponding measured position of the metasurface (blue solid line). We indeed observe that the metasurface is autonomously guided toward the moving source, as indicated by the correlated trends of the two lines (red and blue). This self-guiding behavior is further highlighted by Supplementary Movie 3. Figure 3e shows the top-view photos for three selected time instances: when the radiation source is aligned with the metasurface (middle panel) when the source is horizontally offset to the left by 15° (left panel), and when the radiation source is offset to the right by 15° (right panel). We comment that the actual dynamics of self-guiding will depend both on the metasurface design as well as the profile of the field intensity gradient away from the radiation axis.

These two design features provide direct control over the strength of the guiding effect and can be further tuned depending on the target user. Notably, these results demonstrate metasurface self-guiding along one axis, but the concept is applicable to guiding along multiple directions with an appropriately designed metasurface.

**Acoustic metasurface attractive force**

Finally, we demonstrate the mechanism of attraction, where a metasurface is contactlessly pulled in the direction of the radiation source. As indicated in the schematic in Fig. 1, an object will be pulled if it can efficiently forward scatter the incident wavefront to provide it with additional lateral momentum. Here, we synthesize a metasurface that feels an attractive/pulling force from a source positioned on one side, as shown in Fig. 4a. The fabricated metasurface is shown in Fig. 4a (right) and described in the Supplementary Materials. For scale, a US dime is also shown. Figure 4b plots the simulated lateral force on the metasurface as a function of the source angle, and reveals a region of negative values which denote an attractive force. In the experiment, we measure the displacement of the metasurface as a function of the attack angle of the source θ. In our experimental configuration, a negative metasurface displacement corresponds to the presence of an attractive force. This is indicated by a shaded region in Fig. 4d. For the metasurface in our case, we observe the force is attractive for a range of source attack angles, with the transition from pulling to pushing (negative to positive deflection) occurring at approximately 22°. For attack angles greater than the transition cutoff, the force/deflection becomes positive. The observed metasurface deflection and the presence of the attractive force are generally consistent with the numerical predictions in Fig. 4b. It should be noted that the simulation does not mimic the exact experimental configuration—the simulation is a

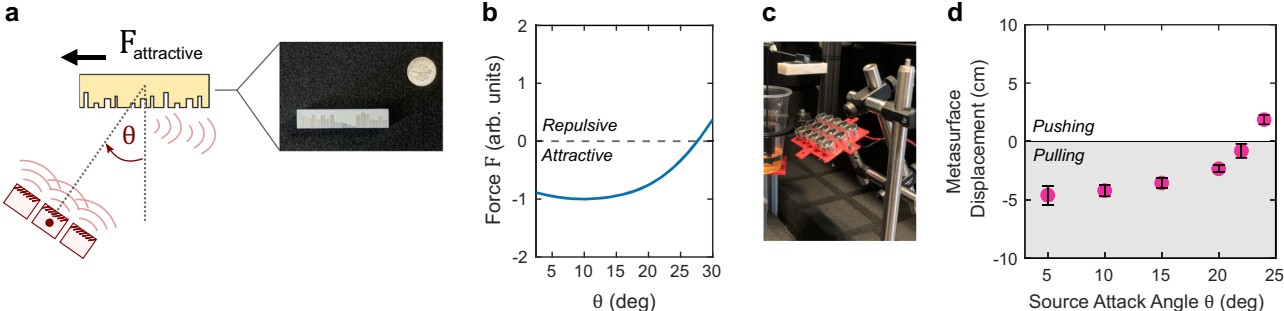

**Fig. 4 | Metasurface attractive forces: concept and demonstration. a** Shown is the configuration of a metasurface designed to experience a force that attracts it toward the radiation source. Right: fabricated metasurface (a dime is shown for scale). **b** Lateral force as a function of the source attack angle. Negative values denote attractive force. The force is extracted from the simulation and normalized to its peak negative magnitude (at -10°). **c** Photo of the metasurface in the setup. **d** Experimentally measured metasurface displacement as a function of the source attack angle $\theta$, as defined in (**a**). Shaded area indicates the condition where a metasurface is pulled toward the source. Error bars represent standard deviation.

two-dimensional characterization of the force on a still/static meta-surface, whereas in the experiment the metasurface freely moves in response to the experienced force. We remark that the acoustic 'pulling' result presented here is primarily intended to showcase the potential of metasurfaces to shape contactless forces; it is likely that stronger pulling behavior could be achieved in more advanced meta-surfaces or in metasurfaces whose design optimization is guided by full three-dimensional wave simulations (as opposed to the limited 2D analysis employed in this work for convenience).

Conceptually, this demonstration shows that an attractive force can be realized on an object whose shape would otherwise not allow this possibility (here, the overall object shape is that of a rectangular slab). This is in contrast to conventional scattering, where attractive force requires the momentum imbalance that can only be achieved in objects of specific conical/tapered shape (e.g., a tapered prism[22]) which can constrain the overall functionality of the object. Meta-surfaces, on the other hand, are not subject to such stringent shape limitations. The metasurface presented here operates in the reflection mode, but a transmissive metasurface could offer a similarly compel-ling tractor beam capability (as schematically shown in Fig. 1d for two independent/non-interfering incident beams).

## Discussion

Our results demonstrate how the modes of contactless acoustic actuation can be tailored by the deliberate subwavelength patterning of the object's surface. Among our main findings, we show that the arrangement of subwavelength surface features can become a pow-erful design degree of freedom to control the acoustic radiation force, independent of the object's shape and size. We further harness this idea to demonstrate example mechanisms of metasurface self-guiding and metasurface tractor beaming. All actuation mechanisms in this work utilized the same block-shaped metasurfaces with the same unit cell shape, which further points to the versatility of the proposed approach. For future studies, it will be interesting to probe actuation dynamics that can be afforded by more complex metasurface morphologies and unit-cell topologies, in both the reflection and transmission modes of operation.

In our demonstrations, the selection of the acoustic frequency beyond the human hearing range has potential benefits for noiseless operation and minimizing disruptions. However, the presented con-cept can be straightforwardly scaled for actuation at other frequencies. There are several potential strategies to enhance the acousto-mechanical metasurface effect. These include increasing the acoustic force by using multiple acoustic sources (e.g., setups comprising dozens/hundreds of transducers[24]), standing wave configurations, reducing the effective sound velocity, as well as designing meta-surfaces and metamaterials that exhibit stronger momentum transfer and/or lower mass. We note that the proposed concept of incorpor-ating metamaterials to control acoustic forces and dynamics is not limited just to large objects and applies equally well to metamaterials that are not sufficiently greater than the wavelength. However, at the intermediate scale (~$\lambda$ to few ~$\lambda$), the design of metamaterial objects might be more numerically driven and may not benefit from an intui-tive description to guide the design. Nevertheless, a potential benefit could be the ability to replicate, or even enhance, the dynamic func-tionality of large metamaterials in a smaller footprint.

We envision a number of compelling extensions and applications of this work. Here, we focused on structures with surface variability along only one direction, and it would be relevant to extend this idea to metasurfaces designed for actuation along multiple axes (e.g., multi-axial guiding). In particular, this could pave the way for concepts such as metasurface levitation, with the potential to overcome the con-straints of conventional levitation. In typical levitation schemes, an object is stabilized and trapped by the acoustic intensity gradient, which works for objects small enough to fit into the nodes of the pressure wave (i.e., smaller than the wavelength). In contrast, we envision a metasurface able to generate its "own" trapping potential to guide and stabilize itself in space, without relying on the profile of the acoustic field. By removing the constraints on object size and shape, such actuation could be relevant for applications in biology, medicine, and robotics, such as contactless assembly and transport, sensing and imaging, and robotic surgery. Applying the same concept in soft and flexible materials[49–51] could enable novel ways to shape and control forces for applications in soft robotics. Our work could also lead to new applications that exploit the frequency-selective nature of meta-surfaces. Unlike conventional surfaces where scattering is insensitive to (small) changes in frequency, a metasurface could generate distinct force profiles based on the wave frequency. Such ability to access and control different dynamics by adjusting the source frequency is not possible in conventional objects and could lead to new mechanisms of multi-modal control, multi-agent robotic manipulation, and actuation applications at the intersection of metamaterials and robotics.

On a conceptual level, the ability to spatially tailor the radiation pressure with subwavelength precision enables the acousto-mechanical response of an object to be designed and realized independently of its dimensions and shape. This feature could be particularly compelling for endowing the object with additional, non-acoustic functionality (e.g., electronic, optical, magnetic). These demonstrations, combined with the ability to create sophisticated acoustic backgrounds and complex metasurface profiles, highlight the

potentially diverse space of mechanisms for contactless actuation shaped by metasurface physics.

## Methods

### Metasurface fabrication and characterization

Metasurfaces in this work were fabricated using photopolymer jetting 3D printing. We found that the sideways orientation of the metasurface during printing yields more consistent printing results and minimizes the chance of material overflow into the grooves of the metasurface. Detailed metasurface dimensions are provided in the Supplementary Information. The experimental setup consists of a suspended vertical tungsten fiber (diameter 75 μm, length 34 cm, shear modulus 160 GPa) that holds a horizontal aluminum rod that acts as a pendulum (length 18 cm), with an overall sensitivity of $1.63 \times 10^{-5}$ N/rad. Supplementary Fig. S1 shows a photo of the setup, where the main axes in the system are defined. Metasurface displacement is characterized by the deflection of the laser spot from laser light that is reflected off the pendulum mirror situated at the base of the fiber. The laser beam path is in the horizontal, $X$–$Y$ plane. While the metasurface is actuated by the acoustic field, a spot-tracing algorithm simultaneously detects the position of the laser spot as inferred from a camera feed of the screen (Supplementary Fig. S1b, c). Subtracting this position from the initial position yields the metasurface displacement, which is plotted on the $y$-axis in Figs. 2e, f, 3d, 4d of the main text.

Acoustic fields are generated from piezoelectric transducers (Manorshi MSO-A1620H). Transducers are connected in parallel and driven by a 20 kHz signal supplied through a driver board. The self-guiding demonstration shown in Fig. 3 utilized a slightly curved transducer array to define a radiation axis that sets the equilibrium position of the metasurface. The schematic of the array holder, with dimensions, is provided in Supplementary Fig. S2. We found there can be significant variability in the acoustic output between different transducers. Additionally, the output of some transducer pieces can vary over time. We characterized the output of the transducers available to us to down-select the most uniform set to comprise the arrays used in this work. For the transducer array used for the results in Fig. 2e, the supplied voltage was 12 V. For the array used for the results in Figs. 3d and 4d, the voltage was 20 V. For all experiments, the surface of the metasurface was 8 cm from the face of the middle transducer of the array being used. Error bars for the plotted experimental data points denote the $\pm 1\sigma$ (standard deviation) confidence interval.

### Metasurface force model

As discussed in the main text, we extract the acoustic force from a finite element approach to calculate the radiation stress tensor over the scattered metasurface acoustic field. We numerically extract the force as the closed contour integral of the second-order acoustic components, i.e., $\mathbf{F} = -\int_S [\langle p_2 \rangle \boldsymbol{n} + \rho_0 \langle (\boldsymbol{n} \cdot \mathbf{v}_1) \mathbf{v}_1 \rangle] dA$, where $\mathbf{v}_1, p_2$ are the first and second-order perturbations in velocity and pressure, respectively, and $\rho_0$ is the density of the medium. As Fig. S3 shows, the integral can be carried out along any fixed surface enclosing the object, as ensured by momentum conservation. For validation purposes, we apply this method on a non-metasurface object (e.g., a deflecting prism) that a metasurface can be arranged to mimic. In Fig. S3b, the acoustic field is incident from the top and the prism is assumed to be larger than the wavelength of sound like a metasurface would be ($L > \lambda$).

In our finite-element method (FEM) analysis, we use the stress tensor approach to extract the lateral $F_l$ and the normal $F_n$ radiation force components as a function of the prism angle $\alpha$. On the other hand, from the nature of the configuration, we develop an analytical force model by approximating the scattering specular reflection for which the momentum transfer math is simplified. Assuming the incident field along the $-z$-axis and the reflection to be specular, it can be shown that the net difference between the scattered and incident field momentum leads to force component expressions given by $F_l \propto$

$\cos(\frac{\pi}{2} - 2\alpha)$ and $F_n \propto -\sin(\frac{\pi}{2} - 2\alpha) - 1$. Figure S3c compares the analytical model (solid lines) with the full finite element (FEM) stress tensor calculation. The FEM analysis considered both an ideal plane wave field (empty circles) and an acoustic field generated by a transducer array (solid circles) to mimic an experimental setting. Units for the force (N/m) stem from the 2D nature of the simulation. Throughout this work, simulations are used to inform the acoustic force trends and, as such, the absolute value of the simulated force (in units of N/m) is not of interest (Figs. 3c and 4b). The absolute force value is linearly proportional to the intensity of the acoustic field specified in the simulation. Computational data for metasurface 1 and 3 (Figs. 2 and 4, respectively) were taken at 8 cm separation between the surface of the metasurface and the face of the transducer array. Computational data from metasurface 2 in Fig. 3 was taken at 12 cm separation to better visualize the direction of the scattered wave.

### Metasurface design

Our strategy for acousto-mechanical metasurface design comprises several steps. First, the profile of the unit cell is selected. We focus on a groove-like metasurface unit cell consisting of a U-shaped element (Supplementary Fig. S4a) because of its simplicity of fabrication. As an example of a different unit cell shape, a space-coiling metasurface is analyzed in the Supplementary. For the U-shaped cell, the dimensions of the ridges/grooves are $u = 0.45$ mm, $v = 0.407$ mm, with the overall unit cell dimension of $a = 3(u + v) = 2.571$ mm. Supplementary Fig. S4b shows the reflected phase profile vs the cell depth $x$ in millimeters ($f = 20$ kHz) and confirms that a full $2\pi$ phase sweep is accessible. Second, for Metasurface 1, we use the phase profile vs cell depth map and the generalized Snell's law to analytically design a deflecting metasurface structure of a given size (45°, $N = 30$ unit cells). The deflection angle was selected based on observations that for our slabs, the lateral force initially increases with the increased angle but eventually becomes weaker for analytical designs at large angles. This analytical structure will represent a starting point for numerical optimization to further increase the force. Third, we refine the metasurface design by treating the $N$ unit cell depths $x_1, \ldots, x_N$ as varying, but independent optimization parameters. We employ a local derivative-free optimization based on the Subplex search algorithm where the objective function is to maximize the lateral force assessed through the stress tensor method. The algorithm was accessed via the MATLAB version of the NLopt package (http://github.com/stevengj/nlopt). The outcome of this process is the final profile in Fig. 2. Notably, the optimized metasurface exhibits 1.8x stronger lateral force than the starting analytical metasurface, which serves as a proof of concept that metasurface acoustic forces can be directly designed for.

Metasurface 2 (Fig. 3, $N = 30$) is a hybrid mirror-symmetric configuration of two analytical steering structures (30°, $N = 15$). To design it, we apply identical considerations and follow the same procedure except we omit the numerical optimization step as there is no unique objective design function for the concept of self-guiding; nevertheless, if a specific profile of the self-guiding behavior is desired in the future, the metasurface could be optimized/refined in the same manner as the first metasurface. Last, the third metasurface (Fig. 4, $N = 20$) seeks to maximize the lateral metasurface pulling force when the transducers are oriented at an angle relative to the metasurface normal (attack angle 20°). We use the same optimization process as before. We note that metasurface 3 is intentionally made slightly smaller ($N = 20$ unit cells) to minimize the number of independent variables and speed-up optimization. All unit cell depth profiles are tabulated and provided in the Supplementary Information. For all designs, the depth $x_i$ is restricted to a maximum of 9 mm.

## Data availability

The data needed to evaluate the conclusions in the paper are present in the paper and/or the Supplementary Information. Additional

supporting data are available at the Data Repository for the University of Minnesota (https://doi.org/10.13020/zkrr-ny94).

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

## Acknowledgements

We thank A. Kumar, R. Rajamani, and M. McAlpine for the helpful discussions. The authors acknowledge support from the University of Minnesota Robotics Institute and the Air Force Office of Scientific Research (AFOSR) under grant number FA9550-22-1-0070.

## Author contributions

M.S., S.K., and O.I. designed the experiments. M.S. carried out the numerical analysis. M.S., S.K. performed the experimental measurements and data analysis. Y.L. assisted with the experimental design. O.I. conceived and supervised the project. All authors contributed to the interpretation of the data and the preparation of the manuscript.

## Competing interests

The authors declare no competing interests.

## Additional information

**Correspondence and requests** for materials should be addressed to Ognjen Ilic.

