## [Peer Review File · Nature Communications]

Reviewer comments, first round review Reviewer #1 (Remarks to the Author):

Reviewer 1:

The applications of acoustic radiation force are widely used for manipulating the target objects such as microparticles and cells in a non-contact way. Generally, the acoustic radiation force is determined by the geometry of the conventional scatterer, which limits the acoustic manipulation. Using an active phased array to control acoustic radiation force has been deeply investigated. But sophisticated electronic circuit systems and high costs hinder its wide applications. Through exquisite structural design, the metasurface (2D metamaterials with subwavelength thickness) could realize controllable reflection, transmission, and extraordinary absorption. These passive methods to manipulate the acoustic radiation force are becoming popular.

In this manuscript, the authors have designed one kind of metasurface which is composed of arranged grooves with different depths in subwavelength scale. Here, three metasurfaces have been proposed to achieve the special functions which are sideways force where the acoustic radiation force is parallel to the object surface, self-guidance where a metasurface object is autonomously guided by an acoustic wave, and tractor beaming where a metasurface object is pulled by the wave. These demonstrations are well designed and fully proved. This paper is well-written with few errors.

Using acoustic metamaterials to manipulate acoustic radiation forces is not new. Acoustic levitation and vortex are reported in many papers. So the significance and importance of this manuscript are my concern despite this work has not been systematically investigated. I hope the authors can provide more valuable practical applications to meet the high standards required by the journal. My specific concerns are as follows.

(1) In the manuscript, the authors used the phrase "contactless forces" in the title to describe the acoustic radiation force. I suggest that the authors might modify the title to directly reflect the essence of force, i.e., acoustic radiation force in order to avoid ambiguity because many non-contact actuation phenomena could be electromagnetic forces.

(2) Common acoustic metasurfaces are composed of Helmholtz resonators, decorated membrane resonators, and coiling-up space structures. In the manuscript, the authors should explain why the groove-like structure was chosen and what its advantages are.

(3) In the demo of the self-guiding metasurface, the authors designed a hybrid mirror-symmetric configuration of two steering structures. If you want to improve the positioning accuracy of the self-guiding, the sound energy should be concentrated, i.e., the sound beam diameter should be as small as possible. However, if the sound beam diameter is so small that it covers several building-block unit cells, the function of the metasurface will deteriorate resulting in weakening the acoustic radiation force, and vice versa. We encourage the author to deeply analyze the relationship between self-guidance and incident beam diameter.

(4) Generally, the acoustic radiation force is very small. Reducing the effective sound velocity can increase the forces. Under the same incident sound intensity, the author might analyze whether the acoustic radiation force produced by metasurface is stronger than that of the conventional acoustic reflector.

(5) Some typos

The picture order is uppercase in each figure, while the article is lowercase.

In the first paragraph of the section "INTRODUCTION", the dashed line in the sentence "which means that the wave force is dictated—and constrained by—the overall shape and size of the object" could be deleted.

In figure 3, Fig.3 (D) appears twice. The latter one should be Fig.3 (E).

Reviewer #2 (Remarks to the Author):

In this paper a target object is fashioned (by 3D printing) into a metamaterial and manipulated using ultrasound. The paper takes as its starting point the recent works on ultrasonic levitation of objects, which are typically small (compared to the wavelength) and simple in geometry (e.g. spheres). The other inputs are metamaterials, or perhaps more correctly, metalayers and metasurfaces. These have the ability to transform input sound fields into a variety of scattered sound fields. The authors link these two ideas together and produce some very exciting new results. The ability of the metamaterial to scatter the incident wave energy in different directions is then used to allow for the application of a near-horizontal force, a stable restoring force to follow a focus as well as an attractive (tractor beam) effect. These ideas are clearly described both in simulation (numerical models) and through experiments. The experiments are particularly elegant. These really are the simplest possible implementations of the ideas which leads to very clear demonstrations of the concepts. The basic concept is novel and the idea will be of significant interest to those working in the fields of acoustics, metamaterials and micromanipulation. Beyond this, the concept that could have implications for other fields such as in vivo robotic surgery.

It would be useful to provide more information about the choice of the metamaterial unit cell and the design process that leads to the metasurfaces. In particular, 1) why choose the "u" shaped unit cell? Would other options work equally well? And 2) could the authors provide some further details as to the metasurface design process that led to the various lateral variations (i.e. $T(x)$). At present this aspect seems to be one of trial and error, is that correct? If possible, it would be useful to provide a clearer description of the design strategy. In this regard the lateral force example is hardest to follow as I would have thought a simple regular diffraction grating (that would reflect a steered beam) would also have a lateral force effect. Yet the profile used here looks much more complex.

The other aspect that is worth exploring in a bit more detail is the size of the metamaterial slab. At present this is said to need to be significantly greater than the wavelength. 3) I am unclear if this really is a design restriction as it is not discussed in much detail. And 4) it would be interesting, if possible, to comment on what would happen if this restriction was relaxed?

Finally, 5) is there any prospect of the metamaterial slab being levitated, i.e. the applied forces being sufficient to overcome gravity?

AUTHOR RESPONSE TO DECISION LETTER
Nature Communications Manuscript ID: NCOMMS-22-14545

We would first like to thank the reviewers for their careful evaluation of our paper and the constructive critique they have provided to us. In this Response Letter, we address each concern raised by the referees and make necessary modifications to the initial submission.

SUMMARY

Reviewer #1 (Remarks to the Author):

The applications of acoustic radiation force are widely used for manipulating the target objects such as microparticles and cells in a non-contact way. Generally, the acoustic radiation force is determined by the geometry of the conventional scatterer, which limits the acoustic manipulation. Using an active phased array to control acoustic radiation force has been deeply investigated. But sophisticated electronic circuit systems and high costs hinder its wide applications. Through exquisite structural design, the metasurface (2D metamaterials with subwavelength thickness) could realize controllable reflection, transmission, and extraordinary absorption. These passive methods to manipulate the acoustic radiation force are becoming popular.

In this manuscript, the authors have designed one kind of metasurface which is composed of arranged grooves with different depths in subwavelength scale. Here, three metasurfaces have been proposed to achieve the special functions which are sideways force where the acoustic radiation force is parallel to the object surface, self-guidance where a metasurface object is autonomously guided by an acoustic wave, and tractor beaming where a metasurface object is pulled by the wave. These demonstrations are well designed and fully proved. This paper is well-written with few errors.

RESPONSE:

We thank the reviewer for the favorable evaluation of the technical quality of the work, and for providing us with many insightful comments. We have carefully revised the manuscript to address the feedback from the reviewer, and below we provide point-by-point responses.

Using acoustic metamaterials to manipulate acoustic radiation forces is not new. Acoustic levitation and vortex are reported in many papers. So the significance and importance of this manuscript are my concern despite this work has not been systematically investigated. I hope the authors can provide more valuable practical applications to meet the high standards required by the journal.

RESPONSE:

We would like to take this opportunity to clarify the novelty of the manuscript. Our work demonstrates that for a given object, encoding a pattern of subwavelength features on its surface (i.e., turning its surface into a metasurface) enables the control of acoustic radiation forces and gives rise to actuation dynamics that have previously been inaccessible. We present the first theory and experiments that show this. Unlike conventional levitation (typically used on

subwavelength objects), the significance of our work is that the motion of an object is, for the first time, controlled not by its morphology or size, but by engineered patterns on its surface. We believe this has important consequences, and the example phenomena that we report (anomalous force, self-guidance, tractor beaming) would otherwise not be possible if the object we use did not have a metasurface. We are not aware of competing reported uses of acoustic metamaterials to control acoustic forces and their dynamics.

We fully agree with the reviewer about the many reports of acoustic levitation, and we tried to cite as much of the relevant literature as possible in our initial submission. While acoustic holograms and vortices can structure the wave field for acoustic levitation, it is worth emphasizing that these only modify the wavefront that is *incident* on the particle/object, but the manipulated object is still subwavelength/spherical (e.g. [14]), which is fundamentally distinct from the physics that we report.

We especially appreciate the reviewer's comment on practical applications, and we added a discussion in the revised manuscript, where we also included application examples suggested to us by Reviewer 2.

CHANGES:

Per the reviewer's suggestion, the revised manuscript now includes a discussion on applications in the "Summary and Outlook" section, specifically Pages 6 (bottom) and 7 (top), highlighted.

(1) In the manuscript, the authors used the phrase "contactless forces" in the title to describe the acoustic radiation force. I suggest that the authors might modify the title to directly reflect the essence of force, i.e., acoustic radiation force in order to avoid ambiguity because many non-contact actuation phenomenons could be electromagnetic forces.

RESPONSE:

We thank the reviewer for this suggestion, and we agree that a title modification will eliminate any ambiguity regarding the nature of the forces discussed in this paper. Specifically, the proposed revised title "Shaping contactless radiation forces through anomalous acoustic scattering" now explicitly states that these are *radiation* forces and that they originate from acoustic scattering.

CHANGES:

Modified the manuscript title to reflect the radiation essence of the force.

(2) Common acoustic metasurfaces are composed of Helmholtz resonators, decorated membrane resonators, and coiling-up space structures. In the manuscript, the authors should explain why the groove-like structure was chosen and what its advantages are.

RESPONSE:

The reviewer raises an important point. The main reason we chose the groove-like unit cell topology was to simplify fabrication. Even though the smallest feature of our unit cell operating

at 20 kHz is within the resolution limits of standard polyjet 3D printing, we wanted to minimize the risk of fabrication errors and the need for, and the cost of, reprinting.

However, the acousto-mechanical metasurface physics and the phenomena that we report are general and not unique to the chosen unit cell design. As a demonstration, in addition to the groove-like metasurface profile, we also investigate an example metasurface composed of a space-coiling unit cell topology. The details of this analysis are provided in the new Supplementary Information section, titled “Alternative metasurface topologies for shaping acoustic radiation forces”. To summarize briefly, we find that a judiciously designed coiled metasurface can perform comparably to a groove-like structure and that it can also be enhanced in a similar fashion through the design optimization that we proposed. We note that such coiled metasurfaces can be particularly relevant for acousto-mechanical actuation in the transmission mode of operation.

CHANGES:

Updated the main text to explain the choice of a grooved-like structure (Page 3, paragraph 2, highlighted). Revised the Supplementary Information, and added a new section where an alternative space-coiling unit cell is designed and analyzed using the same formalism (SI Section “Alternative metasurface topologies for shaping acoustic radiation forces”)

(3) In the demo of the self-guiding metasurface, the authors designed a hybrid mirror-symmetric configuration of two steering structures. If you want to improve the positioning accuracy of the self-guiding, the sound energy should be concentrated, i.e., the sound beam diameter should be as small as possible. However, if the sound beam diameter is so small that it covers several building-block unit cells, the function of the metasurface will deteriorate resulting in weakening the acoustic radiation force, and vice versa. We encourage the author to deeply analyze the relationship between self-guidance and incident beam diameter.

RESPONSE:

We agree that an analysis of the relationship between the beam diameter and self-guidance is warranted. In the revised Supplementary Information, we added a discussion and a new Figure S6 to quantify the effect of the beam diameter on the self-guiding force. The results in the figure are based on a systematic analysis where for each beam diameter the self-guiding force is numerically evaluated for a range of relative displacements between the metasurface center and the beam axis. We observe that an initial increase in the beam diameter helps enhance the acoustic radiation force (as the reviewer points out). However, as the beam diameter becomes broader, the countering contribution from the opposing symmetric end of the metasurface becomes stronger, and the overall self-guiding force is weakened. Because of this, we would expect that the relative off-axis displacement for which the force is the strongest (for a given diameter) would increase with the beam diameter increase, and this trend is indeed observed in our simulations (dashed lines in Fig S6).

CHANGES:

The SI is revised to include a new figure and discussion where the relationship between the beam diameter and the self-guiding force is systematically analyzed through numerical simulations (SI Section “Relationship between beam diameter and self-guiding force”).

(4) Generally, the acoustic radiation force is very small. Reducing the effective sound velocity can increase the forces. Under the same incident sound intensity, the author might analyze whether the acoustic radiation force produced by metasurface is stronger than that of the conventional acoustic reflector.

RESPONSE:

We thank the reviewer for bringing up this insightful point. The primary use of metasurfaces in this work is to offer a mechanism for controlling the direction of acoustic momentum change at the surface of the manipulated object. As such, we do not expect that it would enhance the force relative to a specularly reflecting non-metasurface, i.e., non-patterned specularly reflecting surface. This is corroborated by our numerical COMSOL acoustic simulations. However, the referee brings up a potentially relevant mechanism for further adjusting the intensity of acoustic forces. Motivated by the reviewer's feedback, in the revised manuscript we added a discussion on strategies to enhance the acoustic force.

CHANGES:

Included a discussion on possible strategies to enhance the strength of the acoustic force in the revised manuscript (highlighted text on Page 6, paragraph 2).

(5) Some typos

The picture order is uppercase in each figure, while the article is lowercase.

RESPONSE:

We agree, and we thank the reviewer for catching this. All figures and figure captions now have lowercase labels.

CHANGES:

Labels changed from uppercase to lower case in all figures and figure captions.

In the first paragraph of the section "INTRODUCTION", the dashed line in the sentence "which means that the wave force is dictated—and constrained by—the overall shape and size of the object" could be deleted.

RESPONSE:

We agree. We corrected the sentence per the reviewer's suggestion.

CHANGES:

The revised sentence (Page 1, first paragraph, highlighted text) now reads: "For a typical surface or interface, the transfer of momentum is governed by Snell's law for waves, which means that the wave force is dictated and constrained by the overall shape and size of the object."

In figure 3, Fig.3 (D) appears twice. The latter one should be Fig.3 (E).

RESPONSE:

We thank the reviewer for pointing this out. Figure 3 labels are now fixed.

CHANGES:

Revised Figure 3 has correct labels.

Reviewer #2 (Remarks to the Author):

In this paper a target object is fashioned (by 3D printing) into a metamaterial and manipulated using ultrasound. The paper takes as its starting point the recent works on ultrasonic levitation of objects, which are typically small (compared to the wavelength) and simple in geometry (e.g. spheres). The other inputs are metamaterials, or perhaps more correctly, metalayers and metasurfaces. These have the ability to transform input sound fields into a variety of scattered sound fields. The authors link these two ideas together and produce some very exciting new results. The ability of the metamaterial to scatter the incident wave energy in different directions is then used to allow for the application of a near-horizontal force, a stable restoring force to follow a focus as well as an attractive (tactor beam) effect. These ideas are clearly described both in simulation (numerical models) and through experiments. The experiments are particularly elegant. These really are the simplest possible implementations of the ideas which leads to very clear demonstrations of the concepts.

The basic concept is novel and the idea will be of significant to interest to those working in the fields of acoustics, metamaterials and micromanipulation. Beyond this, the concept that could have implications for other fields such as in vivo robotic surgery.

RESPONSE:

We appreciate the reviewer's positive assessment of the novelty and the impact of the work. We are especially grateful for their suggestions of potential applications that go beyond what we have considered and which we have now incorporated into our revised manuscript (e.g., robotic surgery).

CHANGES:

We included a discussion on applications in the "Summary and Outlook" section of the revised manuscript (Pages 6 (bottom) and 7 (top), highlighted).

It would be useful to provide more information about the choice of the metamaterial unit cell and the design process that leads to the metasurfaces. In particular, 1) why choose the "u" shaped unit cell? Would other options work equally well?

RESPONSE:

The reviewer raises an important point. The main reason we chose the groove-like unit cell topology was to simplify fabrication. Even though the smallest feature of our unit cell operating at 20 kHz is within the resolution limits of standard polyjet 3D printing, we wanted to minimize the risk of fabrication errors and the need for, and the cost of, reprinting.

However, the acousto-mechanical metasurface physics and the phenomena that we report are general and not unique to the chosen unit cell design. As a demonstration, in addition to the groove-like metasurface profile, we also investigate an example metasurface composed of a space-coiling unit cell topology. The details of this analysis are provided in the new Supplementary Information section, titled “Alternative metasurface topologies for shaping acoustic radiation forces”. To summarize briefly, we find that a judiciously designed coiled metasurface can perform comparably to a groove-like structure and that it can also be enhanced in a similar fashion through the design optimization that we proposed. We note that such coiled metasurfaces can be particularly relevant for acousto-mechanical actuation in the transmission mode of operation.

CHANGES:

Updated the main text to explain the choice of a grooved-like structure (Page 3, paragraph 2, highlighted). Revised the Supplementary Information, and added a new section where an alternative space-coiling unit cell is designed and analyzed using the same formalism (SI Section “Alternative metasurface topologies for shaping acoustic radiation forces”)

And 2) could the authors provide some further details as to the metasurface design process that led to the various lateral variations (i.e. $T(x)$). At present this aspect seems to be one of trial and error, is that correct? If possible, it would be useful to provide a clearer description of the design strategy. In this regard the lateral force example is hardest to follow as I would have thought a simple regular diffraction grating (that would reflect a steered beam) would also have a lateral force effect. Yet the profile use here looks much more complex.

RESPONSE:

The modified and reorganized “Methods” section now contains additional details to clarify the design strategy (subsection titled “Metasurface design”). Here, we outline a three-step design process: (i) unit cell selection and phase mapping; (ii) analytical design, informed by unit cell phase-depth relationship and the generalized Snell’s law; (iii) additional numerical refinement through finite element simulations and design optimization to enhance the acoustic force. It is the last step (iii) where the profile of the metasurface becomes more complex, as the reviewer points out. However, this optimization also substantially increases the lateral acoustic force. This is an important point of our paper, namely that one can judiciously design a metasurface for an enhanced force beyond the starting point provided by an analytical design.

CHANGES:

Revised Methods section to include the details behind the design strategy (section “Metasurface design”, page 9, highlighted).

The other aspect that is worth exploring in a bit more detail is the size of the metamaterial slab. At present this is said to need to be significantly greater than the wavelength. 3) I am unclear if this really is a design restriction as it is not discussed in much detail.

RESPONSE:

The reviewer raises an important and insightful point. The concept of incorporating metamaterials/metasurfaces to control acoustic forces and dynamics as introduced by the paper is

indeed not limited to large objects, as the reviewer correctly notes. We focus on the regime where size is sufficiently greater than wavelength because it provides intuitive physics to guide and explain the novel dynamics. Specifically, the link between (i) a metasurface that transforms the momentum of sound and (ii) that momentum transformation resulting in a controllable force is intuitively captured in this regime.

We note that the numerical-experimental approach we introduced to predict and measure acoustic forces would work just as well for metamaterial objects that are not sufficiently greater than the wavelength or are even comparable to the wavelength. An envisioned technical challenge is that the design of such objects might be almost exclusively numerically driven and would require full three-dimensional finite-element simulations. The upside is that we might be able to replicate (and even enhance) the dynamic functionality of large metamaterial slabs in a smaller footprint.

CHANGES:

The revised manuscript now includes a discussion on the size of the metamaterial slab and related implications (Page 6, third paragraph, highlighted). Please also see the response to point 4) below.

And 4) it would be interesting, if possible, to comment on what would happen if this restriction was relaxed?

RESPONSE:

Continuing from point 3 above, shaping acoustic forces and dynamics in the “intermediate” ($\sim\lambda$ to few $\sim\lambda$) metamaterial size range is very interesting. On one hand, such objects would expand the degree of control in applications where existing acoustic manipulators/levitators are already used (and which are fundamentally limited to manipulating subwavelength-sized objects $<\lambda$). On the other hand, they might be able to achieve similar dynamic functionality of substantially larger metamaterial slabs. However, the design of metamaterial objects at this intermediate scale may not benefit from an intuitive physical picture and may need computationally demanding simulations. Nevertheless, the combination of numerical stress tensor analysis and experimental force characterization of our work would remain directly relevant.

CHANGES:

The revised manuscript now includes a discussion on the size of the metamaterial slab and related implications (Page 6, third paragraph, highlighted).

Finally, 5) is there any prospect of the metamaterial slab being levitated, i.e. the applied forces being sufficient to overcome gravity?

RESPONSE:

We thank the reviewer for raising this exciting prospect. Indeed, we are confident this is possible. One strategy is to increase the radiation pressure force through configurations where the sound is generated by many sources. Arrays comprising dozens/hundreds of transducers have been shown to levitate large sphere-like objects [24]. Additionally, a metasurface can be designed with the intent of maximizing momentum transfer & minimizing mass. However, the

most interesting aspect of a levitating metasurface is the ability to “self-balance”. We envision metasurfaces that can achieve levitation stability with respect to translational and rotational (pitch/yaw/roll) displacement. In a sense, a levitating metasurface can be thought of as a conceptual extension of the self-guiding metasurface from Fig. 3, but where the levitation stability stems from the Jacobian matrix of the stiffness coefficients corresponding to translational and rotational perturbations near the equilibrium. The connection between dynamics, stability, and metasurface physics presents a compelling new approach to contactless manipulation of objects.

CHANGES:

Included a discussion on the prospect of increasing acoustic forces and levitating a metamaterial slab in the “Summary and Outlook” section (Page 6, in paragraphs 2 and 3).

Reviewer comments, further review

Reviewer #1 (Remarks to the Author):

The author's answer dissipated my concerns. I would recommend that the article be published in this Journal.

Reviewer #2 (Remarks to the Author):

The authors have addressed the minor points I raised and so I hope to see the paper published in the near future.